# Advances in Silica-Based Large Mode Area and Polarization-Maintaining Photonic Crystal Fiber Research

**DOI:** 10.3390/ma15041558

**Published:** 2022-02-18

**Authors:** Yuan Ma, Rui Wan, Shengwu Li, Liqing Yang, Pengfei Wang

**Affiliations:** 1State Key Laboratory of Transient Optics and Photonics, Xi’an Institute of Optics and Precision Mechanics, Chinese Academy of Sciences (CAS), Xi’an 710119, China; mayuan18@mails.ucas.ac.cn (Y.M.); wanrui17@mails.ucas.ac.cn (R.W.); lishengwu2017@opt.cn (S.L.); yangliqing@opt.ac.cn (L.Y.); 2Center of Materials Science and Optoelectronics Engineering, University of Chinese Academy of Sciences, Beijing 100049, China

**Keywords:** large mode area, polarization-maintaining, photonic crystal fiber, microstructure fiber, high-power fiber lasers, single-mode operation

## Abstract

In recent years, photonic crystal fibers (PCFs) have attracted increasing attention. Compared with traditional optical fibers, PCFs exhibit many unique optical properties and superior performance due to their high degree of structural design freedom. Using large-mode area (LMA) fibers with single-mode operation is essential to overcoming emerging problems as the power of fiber lasers scales up, which can effectively reduce the power density and mitigate the influence of nonlinear effects. With a brief introduction of the concept, classification, light transmission mechanism, basic properties, and theoretical analysis methods of PCFs, this paper mainly compiles the worldwide development of large-mode area and polarization-maintaining (PM) PCFs, and finally proposes possible technical routes to realize the single-mode operation of LMA-PCFs and PM-LMA-PCFs. Finally, the future development prospects of the PCFs are discussed.

## 1. Introduction

Photonic crystal fibers [1,2,3] are also known as microstructured fibers [4] or porous optical fibers [5]. They are a particular type of optical fiber characterized by the periodic arrangement of microstructures around a solid or hollow defective core, forming the fiber’s cladding, as shown in Figure 1. Silica-based PCFs can be divided into two types, i.e., solid-core fibers and hollow-core fibers [6]. The solid-core PCFs are fibers with silica glass capillaries arranged around the core in a periodic pattern, and the refractive index of the core material is higher than that of the cladding material. Hollow-core PCFs are fibers with silica glass capillaries arranged in a periodic pattern around a silica glass tube, and the core area contains an air hole. PCFs’ light-guiding mechanism is greatly distinguished from the total internal reflection (TIR) light-guiding mechanism of conventional optical fibers. 

Based on the different light-guiding mechanisms in solid-core and hollow-core PCFs, we can further categorize PCFs into three main types, i.e., TIR-PCFs, hollow-core photonic bandgap fibers (HC-PBGFs), and hollow-core anti-resonant (HC-AR) PCFs. First, the TIR-PCFs are refractive index guided, which uses a modified total internal reflection principle for light transmission. In other words, light is confined to the higher refractive index region in the solid-core. TIR-PCFs include the dispersion-compensated and dispersion-flat PCFs for dispersion control, and the ones with large mode areas for high-power laser transmission (Figure 2a,b). Those with a high birefringence characteristic (Figure 2c) can also be applied to maintain the polarization states in the PCFs. Figure 2e shows a solid-core large-pitch photonic crystal fiber. This fiber allows for large mode area single-mode operation based on higher-order mode delocalization. PCFs for optical filtering, optical coupling devices, and highly nonlinear PCFs for generating supercontinuum spectra are exemplified in Figure 2a. Secondly, hollow-core photonic bandgap fibers (HC-PBGFs) use the photonic bandgap effect to confine light to an air core that has a lower refractive index than the surrounding region, as depicted in Figure 2f, for which the energy of the optical field is concentrated in the large air hole of the core where the defect is formed, using the defect state to guide light [6]. Photonic bandgap-type PCFs have specific requirements for light waves, and only waves meeting certain wavelengths can appear to transmit in them with the photonic bandgap effect. For the third type, as shown in Figure 2d, the hollow-core anti-resonant (HC-AR) PCFs confine light to the core by suppressing the gap coupling between the light propagating in the core and the outer cladding glass or cladding tube through the light-guiding mechanism of anti-resonant reflection. The light transmission mode is propagated in the core air for both HC-PBGF and HC-AR PCFs, which are unified as hollow-core photonic crystal fibers (HC-PCFs). HC-PCFs, due to their broadband light conduction and ultrahigh laser damage threshold, play essential roles in ultrashort and ultra-intense laser pulse transmission [8], single-cycle pulse generation [9], and low-latency optical communication [10], etc.

## 2. Properties and Theoretical Analysis Methods of PCFs

PCFs have many unique optical properties compared to conventional optical fibers. These include the endlessly single-mode property [17,18,19,20,21,22], low confinement loss [23], adjustable dispersion property [24], high birefringence property [25], large mode area [26,27], and large numerical aperture, which exhibit dominant advantages in overcoming various challenges faced by traditional fiber lasers as laser power scales up rapidly. For instance, the PCFs may achieve a sizable single-mode field area, which can effectively reduce the power density and mitigate the influence of nonlinear effects. It can also improve the fiber material’s damage threshold while ensuring the single-mode transmission quality. As another example, the PCFs can achieve a large numerical aperture of the inner cladding, which helps to improve the optical pumping coupling efficiency and make high-power output possible with a relatively short fiber length.

### 2.1. Endless Single-Mode Transmission Characteristics Analysis

PCFs can be designed to have the endless single-mode property, where only the fundamental mode can propagate through the fiber core for all wavelengths [18]. The cutoff wavelength analysis of PCFs is not as simple as conventional fibers, because all modes propagating in PCFs with a finite number of air hole rings are leaky [28]. An analysis of the single-mode property for different structures of optical fibers requires careful consideration of the most reasonable analysis method to be used.

Conventional refractive index guided fibers usually use the V-parameter to calculate the normalized frequency. When the calculated V ≤ 2.405, it can be judged as having a single-mode operation [6]. The refractive index of the cladding and core in conventional fibers is uniformly distributed and it is easy to calculate the V-parameter and analyze the single-mode property. Another single-mode verification method for conventional fibers is when the confinement loss of the fundamental mode is <0.1 dB/m, while the confinement loss of all higher-order modes is >1 dB/m. Usually, when the confinement loss of the first higher-order mode is >1 dB/m, it can be judged as having a single-mode operation [29].

Compared with conventional double-clad fibers, PCFs are more flexible due to the large freedom of their structural design. Single-mode property analysis can be performed by calculating the V-parameter, Q-parameter, effective area and effective refractive index of the second-order mode, confinement loss, core overlap factor, etc. However, the applicability of each method is different, so it is necessary to analyze different fiber structures and select an appropriate criterion. 

By calculating Q-parameters for different normalized wavelengths λ/Λ and estimating the negative minima at different air hole diameter to pitch ratios (d/Λ), the single-mode operating interval of the PCFs can be obtained. The criterion of confinement loss adopted for single-mode operation judgment in traditional optical fibers also applies to PCFs. However, the confinement loss of PCFs strongly correlates with the number of air hole rings. With the increasing number of air hole rings, the confinement loss can be reduced greatly to 10^−7^~10^−11^ orders of magnitude, which is almost negligible. Hence, when the number of air hole rings in the inner cladding is large, the confinement loss is no longer applicable as a single-mode criterion for PCFs. When the V-parameter method is used, it is necessary to calculate the effective refractive index of the fundamental space-filling mode ($n_{FSM}$) and the guided fundamental mode ($n_{eff}$) in the air hole cladding. 

For different fiber structures, the values of the effective core radius ρ are different. Many different definitions of the effective core radius were proposed for fiber structures with a triangular lattice replacing an air hole, such as 0.5Λ [30], Λ/$\sqrt{3}$ [31], 0.625Λ [32], 0.64Λ [33] and Λ [18,34]. The calculated V ≤ π can be used as a judge for single-mode operation. Currently, the control of the laser output mode is no longer strictly limited by the refractive index difference between the fiber core and the cladding. Modal analysis in terms of the V-parameter has some technical limitations, such as the fact that the definition of the core diameter seems to be nonunified in different studies [30,31,32,33,34]. The mode of the fiber can be finely adjusted by the appropriate design of the fiber cross-section geometry, so that the introduction of the core to the overlap factor can be used to determine the percentage of fundamental modes. Three main methods judging whether one PCF can be in single-mode operation or not are introduced afterwards separately. 

Method One: Q-parameter calculation [6,19,35,36].

The analytical method for the cutoff wavelength can be demonstrated by calculating the Q-parameter, which is evaluated by analyzing the turning point of *α*/*k*_0_ with respect to λ*/Λ. There is a significant negative minimum at λ*/Λ in the calculation of the Q-parameter, and the Q-parameter’s expression is shown in Equation (1) [35], where α is the loss constant, and k_0_ is the wavenumber [19].
(1)$$Q=\frac{d^{2}\log{\left[\frac{\alpha }{k_{0}}\right]}_{\;}^{\;}}{d^{2}\log\left(\Lambda \right)}\;$$

By fixing the wavelength λ, the Q-parameters for different normalized wavelengths λ*/Λ and estimating the negative minima at different d/Λ conditions, the single-mode and multimode operation intervals of the PCFs can be obtained. As shown in Figure 3a, the larger the number of air hole rings, the sharper the turning point of the Q-parameters [36], and the more reliable this analytical method is. It should be emphasized that the Q-parameter method effectively determines the cutoff wavelength only for PCFs with a large number of air apertures.

Method two: Normalized cutoff frequency (V) [18,19,20,35,37].

The V-parameter is easy to calculate in standard fibers, because it depends on the core radius and the refractive index difference, which are well defined. For PCFs, the V-parameter differs from that of standard fibers, and the normalized frequency V is shown in Equation (2) [18,19,20,37]:(2)$$V=\frac{2\pi \rho }{\lambda }\sqrt{n_{eff}^{2}-n_{FSM}^{2}},$$
where $n_{eff}$ is the effective refractive index of the fundamental guided mode, and $n_{FSM}$ is the effective refractive index of the fundamental space-filling mode in the air hole cladding. Usually, Λ is the hole-to-hole distance (also called pitch) of the triangular lattice. For the studied PCFs, a suitable effective core radius needs to be selected, and ρ in the equation represents the effective radius of the fiber core. ρ can be defined as 0.5Λ, Λ/$\sqrt{3}$, 0.625Λ, 0.64Λ, where Λ depends on the fiber structure.

In 2009, Stefano Selleri et al. studied triangular PCFs with different core geometries, with a focus on the cutoff wavelengths of PCFs with one and seven air holes removed [35]. The boundary describing the single-mode and multimode operation regions in the phase diagram of the restless single-mode region was calculated by considering the leakage loss of the second-order mode. As shown in Figure 3b, for the fiber with a core replaced by one air hole, its operation is in single-mode without cutoff when d/Λ < 0.405. For the fiber with its core replaced by seven air holes, single-mode operation is maintained without cutoff when d/Λ < 0.035.

The numerical aperture (NA) of an optical fiber is an essential parameter, which indicates the ability of the end face of the fiber to receive incident light, and the magnitude of its value determines the ability of the fiber to receive light and the effect on mode dispersion. The numerical aperture of PCFs is expressed as Equation (3):(3)$$NA=\sqrt{n_{0}^{2}-n_{eff}^{2}},$$
where $n_{eff}$ is the effective refractive index, which is related to the structural parameters of the cladding. When the aperture of the cladding changes, the effective refractive index $n_{eff}$ also changes, so the NA versus d/Λ curve can also be calculated.

Method Three: Core overlap factor (Γ) [38,39]
(4)$$\Gamma =\iint_{hex}^{\;}i\left(x,y\right)dxdy,$$
(5)$$i\left(x,y\right)=\frac{1}{P}Re\left[\frac{E\times H^{*}}{2}\cdot \hat{z}\right].$$

The expression of the overlap integral is shown in Equation (4), and hex is the cross-section of the core’s hexagonal fiber core doping. In Equation (5), $i\left(x,y\right)$ is the normalized intensity distribution of the guided mode [38,39], where P is the integral of the intensity over the entire fiber cross-section. The fundamental mode intensity distribution is calculated from the Poynting vector definition, which involves the three components of both the electric and the magnetic fields of the guided mode. The overlap integral is an important parameter to describe the single-mode property of ytterbium (Yb)-doped PCFs with a large mode area. It provides the interaction between the Yb ions and the optical signal as well as describing how tightly the modes are confined in the fiber core, helping to distinguish whether the modes are guided or not. The concept of normalized intensity is derived from the definition of the Poynting vector, which is the energy flow density vector in the electromagnetic field, so the dual integral over the core and the entire cross-section can be calculated using the time-averaged energy density. First, the magnetic field *H* = (*Hx*, *Hy*, *Hz*) on the fiber cross-section is calculated by finite element analysis, and then the electric field *E* = (*Ex*, *Ey*, *Ez*) is obtained by Maxwell’s equations based on the distribution of the magnetic field. Finally, the core overlap factor (Γ) is obtained by double integration of the normalized intensity of the fiber core region.

In 2012, Mette Marie Jørgensen et al. proposed a single-mode determination criterion when optimizing a distributed mode-filtered fiber amplifier [38]. This determination criterion is based on the evaluation of the mode overlap in the doped core region, defined as the normalized integral of the mode intensity over the core region. To obtain effective SM operation by means of a differential amplification mechanism, it defines the fundamental mode (FM) overlap integral as higher than 0.8, and the overlap difference between FM and the most detrimental higher-order mode (HOM) should be >0.25, which can be used as a criterion for single-mode operation in optical fibers. This method is more suitable for analyzing the structure of a fiber core replacing multiple air holes than the V-parameter method, and the calculation results are more accurate.

### 2.2. Mode Area (A_eff_) and Mode Field Diameter (MFD)

The main parameters affecting the transmission characteristics of PCFs include the air hole diameter (d) and pitch (Λ), fiber diameter (D), and core diameter (D_core_). Therefore, by adjusting the air hole size in the cladding, the refractive index difference (∆*n*) between the cladding and the core of the PCFs can be precisely controlled to reach a minimal value, thus achieving endlessly single-mode operation. At the same time, ∆*n* reduction can also effectively increase the mode field area [17,40]. The mode field area of a PCF is expressed as Equation (6):(6)$$A_{eff}=\frac{{\left(\int \int \left|E^{2}\right|dxdy\right)}^{2}}{\int \int \left|E^{4}\right|dxdy}.$$

E in the formula is the transverse electric field component of the PCFs. The wider the transverse electric field distribution of the cross-section is, the larger the mode field area, which is an essential guide for designing large mode area (LMA) PCFs. The effective mode field diameter (MFD) is expressed as Equation (7):(7)$$MFD=2\sqrt{\frac{A_{eff}}{\pi }}.$$

### 2.3. Fundamental Mode Confinement Loss

Loss is a significant factor that must be considered for optical fibers. The loss of conventional optical fibers has been reduced over the last 30 years. The minimum loss of step index fused silica fiber at 1550 nm can be reduced as low as 0.017 dB/km [18,34]. For both solid-core and hollow-core PCFs, the leakage loss or confinement loss needs to be considered. This loss arises due to the limited number of air holes in the fiber cross-section, and therefore, all PCFs guiding modes are leaky. For example, in a solid-core PCF, the light is confined to the core by air holes, and if the air holes do not provide strong confinement, the light leaks out. The expression for the confinement loss of PCFs [25] is shown in Equation (8):(8)$$C_{loss}=\frac{40\pi }{\ln\left(10\right)\lambda }Im\left(n_{eff}\right)\left[\frac{\mathrm{dB}}{m}\right],$$
where $n_{eff}$ is the effective refractive index of the fundamental mode and $Im\left(n_{eff}\right)$ refers to the imaginary part of the effective refractive index of the fundamental mode. The unit of limiting loss is usually dB/m. According to the mode selection theory of PCFs. Single-mode operation can be achieved in PCFs when the fundamental mode loss is <0.1 dB/m, and all higher-order mode loss is >1 dB/m [29]. The first higher-order mode has the lowest confinement loss of all higher-order modes. It is only necessary to determine whether the confinement loss of the first higher-order mode is >1 dB/m to determine whether all higher-order modes are >1 dB/m. Therefore, the fiber is judged to achieve effective single-mode operation if the requirements are met. When the number of cladding air hole rings is small for PCFs, the fiber single-mode operation can be judged by the fundamental mode and the first higher-order mode confinement loss. However, as the number of cladding air hole rings increases further, the confinement loss decreases sharply. It is impossible to use confinement loss as a single-mode criterion for PCFs.

### 2.4. Dispersion Property

Dispersion is a physical phenomenon that causes pulse spreading in fiber optic transmission due to different pulse frequencies or propagation speeds of various mode components [24]. Dispersion impacts the transmission capacity and distance of the fiber. Dispersion can be regulated by the fiber material, fiber structure, and light transmission mode. Materials with negative dispersion properties can counteract dispersion. Microstructured fibers can also be designed to tailor dispersion. The dispersion is controlled by changing the size and distribution of the air holes in the fiber cross-section. Among the methods of transmitting light, people use optical solitons to transmit light in optical fibers, which can maintain the same shape and speed for a long time and transmit a long distance.

The PCFs dispersion can be tailored to a large degree of freedom due to the flexibility to adjust the fiber microstructure and the high refractive index difference between the silica substrate ($n_{eff}$ = 1.45) and the air hole ($n_{eff}$ = 1.0). PCFs can obtain a wider dispersion range than standard optical fibers. 

PCF dispersion mainly consists of material dispersion (*D_m_*) and waveguide dispersion (*D_w_*). Different materials have different refractive indices. The light of different wavelengths transmits at different speeds in the fiber material, resulting in dispersion production, which causes pulse spreading. Since most PCFs are made of the same type of pure silica or modified silica-based materials, the material dispersion is the same for different structures of PCFs. The material dispersion can be calculated by Sellmeier’s formula (Equation (9)):(9)$$n\left(\lambda \right)=\sqrt{1+\underset{i\geq 1}{\sum }\frac{A_{i}\lambda^{2}}{\lambda^{2}-B_{i}}},$$
where *A_i_* and *B_i_* are Sellmeier coefficients, and *λ* is the wavelength (in μm).

The waveguide dispersion of PCFs is closely related to their microstructure, which can be varied by changing the cross-section structure to obtain different waveguide dispersions by means of Equation (10):(10)$$D_{W}=-\frac{\lambda }{c}\frac{d^{2}n_{eff}^{\;}}{d\lambda^{2}},$$
where $n_{eff}$ is the effective refractive index and *λ* is the transmission wavelength (in μm). To obtain the waveguide dispersion, the effective refractive index versus wavelength should be found first. The effective refractive indices of different modes of PCFs at different wavelengths can be obtained using COMSOL simulation software.

### 2.5. Birefringence Property

There are two orthogonal fundamental modes, *HEx* and *HEy* in a normal single-mode fiber. The propagation constants of these two modes are assumed to be the same, i.e., *β_x_* − *β_y_* = 0. In this case, the linearly polarized light transmitted in the fiber can keep the polarization direction unchanged [26,41,42,43], and the fiber is regarded to have polarization-maintaining capability. The ideal single-mode fiber can be assumed to have an ideal circular cross-section with a circularly symmetric distribution of the refractive index. However, in practice, the core will cause anisotropy of the refractive index in different directions due to external reasons, so that the two base molds polarized in the *x*-direction and y-direction have different propagation constants (*β_x_* ≠ *β_y_*). In fact, *β_x_* − *β_y_* = 0 does not exist. For instance, fiber coupling will inevitably introduce deformation and stress, thus introducing birefringence, which leads to a change in the fundamental mode’s polarization state along with the fiber axially. Periodic polarization mainly changes from linearly polarized light through elliptically and circularly polarized light and back to linearly polarized light.

The vital parameter *B* describes the birefringence property of an optical fiber. It is defined as the difference between the effective refractive indices of the two orthogonal polarization modes, which directly reflects the magnitude of birefringence in a single-mode fiber, as expressed in Equation (11):(11)$$B=n_{x}-n_{y}=∆n_{eff}=\frac{∆\beta }{k_{0}}.$$
where ∆$n_{eff}$ is the difference between the refractive indices corresponding to the two orthogonally polarized polarization modes. For linear polarization, ∆*β* is the difference in the propagation constants of the two linear polarization modes polarized along the slow and fast axes; for circular polarization, ∆*β* is the difference in propagation constants between right-handed and left-handed circularly polarized waves. The birefringence of highly birefringence fibers [6] is approximately in the order of 10^−4^. For low-birefringence fiber, the birefringence *B* is ~10^−9^, while for a conventional single-mode fiber, *B* is about 10^−5^~10^−6^.

The stress in material changes the refractive index of the fiber through photo-elastic effect. This causes birefringence in the waveguide device [44] and affects its performance. Equations (12)–(14) describes the change in refractive index due to the photo-elastic effect:(12)$$N_{x}=N_{0}-B_{1}\times \sigma_{x}-B_{2}\times \left(\sigma_{z}+\sigma_{y}\right),$$
(13)$$N_{y}=N_{0}-B_{1}\times \sigma_{y}-B_{2}\times \left(\sigma_{x}+\sigma_{z}\right),$$
(14)$$N_{z}=N_{0}-B_{1}\times \sigma_{z}-B_{2}\times \left(\sigma_{x}+\sigma_{y}\right),$$
where *B*_1_ and, *B*_2_ are the first and second stress-optic coefficients, respectively. *σ_x_* and *σ_y_* are the positive stress in the *x*- and *y*-axis directions, respectively. *N*_0_ is the refractive index of the unstressed material. The anisotropic variation of the refractive index will lead to a higher birefringence value due to the fiber’s stress-optical effect [44]. The birefringence values *B_s_* is expressed as Equation (15):(15)$$B_{s}=N_{y}-N_{x}=\left(B_{2}-B_{1}\;\right)\left(\sigma_{y}-\sigma_{x}\right).$$

## 3. Research Progress in PCFs

One of the major technological achievements of the twentieth century was the development of a low-loss silica fiber [45,46] with a transmission loss of only 20 dB/km using a modified chemical phase deposition (MCVD) method by Corning Incorporated in the 1970s. From then on, fiberoptics has become an important part of the global communication network. In addition to optical communications, medical devices, sensors, fiber lasers, laser machining and manufacturing, and other fields, have all developed rapidly due to emerging novel fiberoptics. With the fast development of science and technology, the performance requirements of traditional optical fibers have gradually increased. 

In 1992, Philip Russell proposed the concept of photonic crystals in optical fibers for the first time [47], by introducing two-dimensional photonic crystals in the fiber cladding that are close in size to the wavelength of light, where the fiber core has a defective structure missing an air hole and light is restricted to propagate in the fiber core. In 1996, J.C. Knight fabricated the first refractive index guided photonic crystal fiber with a solid-core [48]. In 1999, he produced the first bandgap-type hollow-core photonic crystal fiber for light transmission in air. The successful manufacture of HC-PCF enriched the theory of photonic crystals and further opened the door to the development of large-mode area PCFs.

With the rapid development of high-power fiber lasers, the further increase in laser output power is greatly limited by the nonlinear effect, which results in laser-induced optical, thermal, and mechanical damage to the fiber. The nonlinear effect can be prohibited by increasing the fiber’s mode field area, i.e., the larger the mode field area, the weaker the nonlinear effect. Therefore, the design of an LMA-PCF can solve the problem of nonlinear effect-induced fiber damage in the power scaling of fiber lasers [49].

### 3.1. Progress of LMA-PCFs 

Since 1995, research on large-mode field PCFs has progressed rapidly. Figure 4 shows the landmark research advances in single-mode (SM) LMA-PCFs and polarization-maintaining (PM) LMA-PCFs from 1996 to 2021, and their detailed fiber structure parameters, mold field diameters and PM performance are summarized in Table 1. In Figure 4, the blue points in the figure show the development of the mode field diameter of refractive index guided PCFs. The orange points show the development of PM-LMA-PCFs, for which the birefringence reaches a magnitude of 1 × 10^−4^. Currently, the world’s leading institutions studying PCFs include the University of Bath, the University of Southampton, the University of Jena, IRMA (USA), the US Air Force Research Laboratory (AFRL), NKT and Crystal Fiber in Denmark, NTT Laboratories in Japan, and the Beijing University of Technology, Huazhong University of Science and Technology, National University of Defense Technology, and Shanghai Institute of Optics and Fine Mechanics (SIOM), in China. 

The first PCF (Figure 5a) was developed in 1998 by J.C. Knight et al. at the University of Bath, UK [51]. Its mode field diameter is only 22 µm and the mode field area is 380 µm^2^. In 2004, a large-core diameter single transverse-mode Yb-doped PCF with an increased mode field diameter of 35 µm and mode field area of approximately 1000 µm^2^ was designed at the University of Jena, Germany [55]. As shown in Figure 5b, seven air holes are replaced in the fiber core and the core section is doped with Yb ions. The fiber has a core diameter of 40 µm, an inner cladding diameter of 170 µm and outer cladding diameter of 590 µm, an air hole diameter of d = 1.1 µm (four air hole rings) and a pitch of 12.3 µm, and a numerical aperture of 0.03. In experimental tests, this fiber was used to amplify a 10 ps pulse to a peak power of 60 kW. In 2005, the University of Jena reported a novel Yb-doped fiber (Figure 5c) that combines the advantages of rod fiber and fiber gain media [54]. It has the external dimensions of a rod with a fiber diameter of a few millimeters. The fiber has a core diameter of 35 μm, a cladding diameter of 117–141 μm, d/Λ = 0.33, and a mode field diameter of 30 μm. 

In 2006, a Yb-doped rod PCF with a core diameter of 100 μm and an output mode field diameter of 85 μm (Figure 5d) was reported by U.S. Air Force Research Laboratory [67]. The rod PCF, as an amplifier master oscillator, gave a peak power of 4.5 MW, corresponding to a pulse energy of 4.3 mJ and an average power of 42 W, with a slope efficiency of 60%, and an M^2^ of 1.3. In the same year, another Yb-doped PCF was reported by the University of Jena [61]. This PCF’s structure is shown in Figure 5e, where the cladding consists of a triangular hole structure with d/Λ = 0.19, and the core consists of a part missing 19 air holes. The corresponding core diameter is 60 μm, and the fundamental mode field diameter reaches 50 μm. This fiber has low nonlinearity and can amplify short laser pulses to very high peak power. In 2008, Crystal Fiber A/S, Denmark, reported a Yb-doped single transverse mode rod-type PCF [62] that combines low nonlinearity and polarization-maintaining property. Its structure is shown in Figure 5f, this structure obtains high birefringence values by introducing two stress regions with higher refractive indices and different thermal expansion coefficients in the photonic crystal cladding. The mode field area of the fundamental mode is up to 2300 μm^2^ with a mode field diameter of approximately 54 μm. With this fiber, a polarization of >85% up to 163 W output power was obtained without any polarization element inside the cavity. The beam quality is improved with M^2^ reduced to 1.2, compared with that of the Yb-doped rod PCF reported by AFRL [67]. The single-polarization wavelength range is 1030~1080 nm, which has a good overlap with the gain band of the Yb-doped silica fiber. 

In 2011, NKT Denmark designed a large mode area PCF [64] with a higher-order mode filter structure. As shown in Figure 5g, the cladding structure of this fiber is not only a two-dimensional photonic crystal formed by air holes, but it also introduces a high refractive index ring structure at intervals around the air holes, which is resonantly coupled to the higher-order mode of the fiber core and acts as a spatially distributed mode filter (DMF). The fiber core consists of Yb-doped silica replacing the missing 19 air holes, with a core diameter of 85 µm, a hole-to-hole distance of 14.5 µm, and d/Λ as a variable. Single-mode characteristics are demonstrated in this Yb-doped rod fiber with a length of 50 cm by filtering the higher-order modes with the addition of a DMF. This fiber amplifier’s fundamental mode field diameter at 1064 nm is 59 µm, and the pump absorption at 976 nm is 27 dB/m. In 2011, the University of Brussels, Belgium, reported a bendable PCF for short pulse high-power fiber laser application [59]. This fiber has a dual lattice structure, which is achieved by introducing air holes with different air hole and hole pitch ratios in the cladding, as shown in Figure 5h. It has a single-mode mode field area of 1454 μm^2^ without bending and a mode area of 655 μm^2^ in a fiber with a bend radius of 10 cm. The bending loss in the higher-order mode is greater than 50 dB/m, while the bending loss in the fundamental mode is less than 0.01 dB/m, thus enhancing the single-mode operation.

In 2012, the Optics and Electronics Laboratory (Fujikura Japan) investigated a highly efficient single-mode all-solid-state PCF [53] for a high-power fiber laser with a large effective area and low bending loss. The fiber core diameter is 48 μm, and the cladding consists of five layers of high refractive index germanium-doped silica, which achieves a mode field area of 712 μm^2^ in the first bandgap, a mode field diameter of 30 μm with a beam quality factor M^2^ of 1.05, close to the diffraction limit. Its fundamental mode loss is <0.1 dB/m. In 2017, the Shanghai Institute of Optics and Fine Mechanics (SIOM) and the German Institute of Microstructure Technology (GIMT) jointly investigated a method to control Al/F doping in silica glass by controlling the Yb doping combination through Al^3+^/F^–^/P^5+^ co-doping. A modified sol-gel method is used to prepare the low-refractive index silica core glass preform. This glass preform maintains excellent optical homogeneity (refractive index variation < 2.6 × 10^−4^) and spectral properties of Yb^3+^ suitable for fabricating high-power LMA-PCF amplifiers [60]. Finally, a large mode area fiber with a core diameter of 50 µm was obtained by the stacked capillary drawing method. The fiber has single-mode operation with a beam quality factor M^2^ = 1.4. In the 1030 nm pulsed amplified laser experiment, an average amplification peak power of 97 W and an optical efficiency of 54% were obtained for a 6.5 m long fiber. In 2021, the French National Center for Scientific Research (CNRS) reported a Yb-doped core PCF with a unique cladding structure [56]. The preform core was also prepared using the sol-gel method, and the structure effectively suppressed higher-order modes, allowing the fiber to transmit a single-mode. The mode field diameter reaches 35 μm, with a beam quality factor M^2^ =1.18 and a polarization extinction ratio of up to 17 dB. An average power of more than 90 W can be obtained through experimental tests at a wavelength of 1030 nm with a slope efficiency of 75%.

### 3.2. Progress of PM-LMA-PCFs 

To simplify the polarization control device in laser systems, the PCFs with both a large mode area and polarization-maintaining properties are in great and urgent demand to provide a competitive solution, compared with conventional polarization-maintaining fibers. The latter realizes polarization-maintaining properties through introducing built-in stress elements (such as PANDA, bow-tie and elliptical-stress-layer fibers) or incorporating noncircular cores. In contrast, PCFs have greater flexibility to obtain a large mode field area, high birefringence, and adjustable dispersion by structural design. Their potential advantages in terms of transmission characteristics, structural feasibility, cost-effectiveness, and operating band expansion have laid the foundation for the renewal of polarization-maintaining fibers. The development of large mode field polarization-maintaining PCFs has been greatly accelerated by worldwide research.

In 2000, the first highly birefringent PCF which has a birefringence value of 3.7 × 10^−3^ at 1550 nm and a beat length of 0.41 mm was successfully developed by Ortigosa-Blanch et al. at the University of Bath [41]. In 2005, T. Schreiber et al. at the University of Jena reported the structural design of a single-polarization, single transverse mode LMA-PCF [52] by adding a refractive index-matched stress-applied element to the photonic cladding (Figure 6a). A single-polarization window of ultra-broadband 750~1250 nm was obtained at a bending radius of 1.4 m, while maintaining a large mode area of approximately 700 µm^2^. For high-power operation in a highly polarized laser, an extinction ratio of 15.5 dB and output power of up to 25 W were obtained. In 2007, Mingyang Chen et al. of Jiangsu University proposed a novel highly birefringent large-mode area fiber (Figure 6b) [68]. Birefringence in the fiber was achieved by introducing an anisotropic microstructured core, which consists of an upper doped silica background and a lower doped silica rod. Numerical studies showed that the fiber can achieve a high birefringence in the order of 2 × 10^−4^ with a mode field area of >300 μm^2^.

In 2008, Grzegorz Golojuch at the Wroclaw University of Technology (WRUT), induced mode birefringence by incorporating several small holes in the central region of the core, breaking the hexagonal symmetry of the fiber cross-section (Figure 7a) [69]. The fiber has a mode field diameter of 10 μm and a cutoff wavelength of 1.3 µm, and the birefringence reaches 1 × 10^−4^ at 1300 nm and 1.5 × 10^−4^ at 1500 nm. In the same year [62], the University of Jena and Crystal Fiber A/S jointly reported a Yb-doped single transverse-mode rod-type PCF that combines low nonlinearity and polarization-maintaining property (Figure 7d), whose mode field diameter reaches 54 µm. In 2013, NKT developed a fiber (DC-200/40-PZ-Yb) consisting of a 40 µm Yb-doped core, a 200 µm pump cladding, and a stress polarization-maintaining structure (Figure 7b) [57,58]. The stress zone consists of 13 rods on the left and right sides of the core, and the stress zone is separated from the core by a layer of air hole rings. The outer side of the stress zone has no air holes and is connected to the base material. This PM-LMA-PCF can realize high polarization performance with an extinction ratio of 25–30 dB. In 2015, Clemson University and Air Force Research Laboratory co-developed an all-solid-state Yb-doped polarizing PCF for single-polarization, single-mode operation (Figure 7c). Three low index boron-doped stress rods on both sides of the fiber core produced birefringence. These boron rods also provided light confinement through total internal reflection. It had an effective mode area of approximately 1150 µm^2^ and a polarization extinction ratio of up to 21 dB [11].

In 2019, NKT Photonics reported a double-clad Yb-doped polarization-maintaining fiber called “DC-250/30-PM-Yb-FUD” with a core diameter of approximately 30 µm (mode field diameter, 24 µm) and 250 µm inner cladding consisted of silica and doped silica [70]. The stress zone consists of hexagons of doped silica rods on both sides of the fiber core, with each hexagon consisting of 30 rods. The polarization is generated by inducing stress-related birefringence in the core that produces a polarization extinction ratio > 18 dB. In the same year, Yangze Soton Laser CO. (OYSL) in China also successfully developed a large mode area Yb-doped fiber combined with polarization-maintaining ability, which has a core diameter of 40 µm and the fiber structure is very similar to the NKT’s PM-LMA-PCF (Figure 7b). It has excellent single-mode characteristics. Very recently, Yangze Soton Laser CO. successfully realized a breakthrough in rod-type PM-LMA-PCF [65]. The fiber’s structure and performance are akin to NKT’s DC-200/40-PZ-Yb, whose core diameter reaches 85 μm with a mode field diameter of 65 μm and mode field area > 3000 μm^2^, and it can maintain single-mode operation at 1030 nm band with a lower beam quality factor (M^2^ < 1.1). This fiber’s polarization extinction ratio was up to 18 dB.

In summary, significant breakthroughs in the structure design, preform preparation and fiber fabrication technology for PCFs have been made, which have overall contributed to performance improvement. This is conducive to the development of high-power laser and femtosecond lasers with the presence of LMA-PCFs and PM-LMA-PCFs, which have a broad application potential in scientific lab research, industrial laser processing, national defense, and other fields. Until now, the largest mode field diameter that has been achieved is 75 μm in the DC-285/100-PM-Yb-Rod rod-type PCFs by NKT in 2011 [66], whose structure is proprietary. This rod-type PCF disappeared from the market in 2015. It is noted that China’s technology in LMA-PCFs has developed in leaps and bounds, e.g., LMA-PCFs and PM-LMA-PCFs products have appeared and realized wide industrial applications in recent years. However, there is still a big gap between China and its international counterparts in terms of detailed fiberoptics technology and products. 

## 4. Technical Path to Large Mode Area and Polarization-Maintaining PCFs

The most advantageous property of LMA-PCFs is adjustable dispersion, flexible and adjustable numerical aperture, and compact structure, especially with the ability to reduce nonlinear effects by significantly reducing the unit power density loaded on the fiber cross-section, and thus increase laser damage thresholds. These make LMA-PCFs a new research hotspot in fiberoptics for high-power laser applications. 

The technical routes to achieve a large field area in a single-mode, as summarized in Figure 8, include four main categories. The first category is optimizing the fiber material property. High-order modes can be suppressed by doping the core or inner cladding to control the doping distribution and reduce the high-order mode gain. In 2016, the University of Limoges designed a single-mode microstructured fiber with a large mode area highly doped with rare earth ions [71]. The structure consists of a microstructured cladding formed by four layers of high refractive index germanium-doped silica rods surrounding a hexagonal rare earth-doped silica core. The cladding rods have a parabolic refractive index distribution. The cladding hole spacing is 10 μm, the core area is approximately 500 μm^2^, and single-mode operation can be achieved. In 2003, Stanford University [72] proposed a new fiber called the gain-guided, index anti-guided (GG-IAG) fiber, which has a large diameter core with a negative refractive index step from the cladding to the core, combined with a large enough gain coefficient in the core. Large gain coefficients with very large mode areas are very favorable for robust single transverse mode operation.

The second category is the design of fiber microstructures. The primary method introduces a two-dimensional photonic crystal structure into the cladding. The size of the air holes in the cladding and the hole-to-hole distance were adjusted to regulate the refractive index difference between the cladding and the core to achieve a low-refractive index difference. Part III of this paper describes the major research advances in LMA-PCFs. In addition to introducing two-dimensional photonic crystal structures in the cladding, leaky channel fibers (LCFs) with large air holes or large-pitch fibers (LPFs) with large hole-to-hole distance can be designed to filter out higher-order modes using air holes to form a mode sieve. A single-mode and a large mode area can also be achieved by designing air core fibers with anti-resonant cladding and microstructured core structures. In 2007, IMRA designed a leaky channel fiber [28,73]. This type of fiber that can be precisely designed to produce large leakage losses in higher-order modes while maintaining negligible transmission losses in the fundamental mode. As shown in Figure 9b, the passive leaky channel fiber has a mode field area of 1417 µm^2^ and a mode field diameter of 42.5 µm in the 1064 nm band, and the fundamental mode loss is 0.17 dB/m at 1064 nm. The Yb-doped leaky channel fiber is shown in Figure 9c, it has a calculated mode field diameter of 63.4 µm and a mode field area of 3160 µm^2^ in the 980 nm band. In 2009, IMRA reported a new design [51] of an all-solid-state large mode area fiber for high-power Yb-doped fiber lasers and amplifiers. It consists of a single ring of down doped silicon rods around a seven-cell pure silicon core (Figure 9a). This structure achieves both single-mode operation and low bending loss. These fibers have a near parabolic refractive index distribution with a slight refractive index difference between the core and cladding, capable of reaching ∆n ~ 6 × 10^−5^. Stress-induced PCFs have a tiny refractive index difference between the core and the low-refractive index channel, allowing for a greater increase in core diameter while maintaining single-mode operation. However, refractive index accuracy control is highly demanding and challenging for the fabrication of LMA-PCFs.

The third method that can be used is introducing a discriminating mechanism for higher-order modes. The mode matching technique excites only the fundamental mode of the fiber by injecting seed radiation, and this method can be used for all types of few-mode fibers. However, as the mode field area increases, single-mode operation becomes difficult. The bend-selective mode is a typical single-mode operation that uses the difference in bend loss between the fundamental mode and higher-order modes to achieve higher-order mode filtering. A new method for obtaining single transverse mode operation in multimode fiber amplifiers was reported in 2000 by the U.S. Naval Research Laboratory [82,83]. A 6-m gain fiber with a core diameter of 25 μm was wound around a cylindrical shaft to produce significant bend loss for all modes except the fundamental mode, with the bend loss acting as a distributed spatial filter. A laser amplification efficiency of 64% is obtained, and a single-mode output is achieved with a beam quality factor M^2^ ≈ 1.09. It is also possible to design resonant filtering modules, which can be achieved by a cladding structure with a characteristic solution that matches the effective refractive index of the higher-order modes. Designing resonant filter structures is very challenging. In 2011, the NKT company designed a large mode area PCF [64] with a higher-order mode filter structure, as shown in Figure 5g, which allows resonant coupling of higher-order modes from the fiber core and acts as a spatially distributed mode filter (DMF). A final alternative is to introduce modal sieve structures. For example, the leaky channel fiber (LCF) is a higher-order modal sieve structure. It is also possible to combine several approaches in one fiber. For example, in 2021, Pu Zhou of the National University of Defense Technology, China, developed an all-solid photonic bandgap fiber that considers multi-resonant coupling and leakage channels [84]. The fiber core comprises seven silica rods instead of seven germanium rods. The microstructured cladding comprises four layers of germanium-doped high-refractive index silica rods arranged in a specific order in a low-refractive index background material. The concept of multiple leakage channels and multiple resonant coupling is introduced in the cladding. The dependence of the fiber on the bending direction is eliminated by designing the position of the germanium rod distribution so that the bending loss between the fundamental mode and the higher-order modes has a high loss ratio. At a bending radius of 45 cm, a mode field area of >900 μm^2^ and a loss ratio of approximately 495 was obtained.

The fourth category is mode conversion, which can be achieved using higher-order mode transmission or coupling the higher-order mode of the core into other cores of the cladding. In 2014, the University of Michigan reported progress in increasing the core size of effective single-mode chiral coupled core (CCC) Ge-doped and Yb-doped double-clad fibers to the range of 55 to 60 μm. It experimentally demonstrated its robust single-mode performance [74].

When one designs a fiber with a large mode area, it is first necessary to determine the range of the fiber mode field diameter. The field diameter limit of a traditional single-mode step fiber mode is approximately 15 μm. To further increase mode area, one can use discriminatory mechanisms that include higher-order modes in the fiber. The identification mechanism of higher-order modes mainly includes mode matching, bending selection mode, resonant filter mode, etc. These methods make it possible to achieve a mode field diameter of approximately 15~50 μm. A fiber with such a diameter size can be called a large mode area (LMA) fiber. When introducing pattern-matching techniques into very large pattern areas, they need to be combined with other pattern filtering strategies. Moreover, single-mode implementation through mode matching is usually insufficient for high-power fiber lasers and amplifiers. When the mode field diameter exceeds 50 μm, it can be called a very large mode area (VLMA), and single-mode operation can be achieved by introducing microstructured fibers [50]. 

However, very stringent structure and refractive index selection is required due to the large mode area [12]. This can also be achieved by introducing the concept of mode sieves through a large-pitch fiber, where higher-order modes leak out of the large-pitch cladding. This can also be referred to as leaky cladding, which is not usually present in high-power pumped double cladding fibers. Therefore, rod-shaped double-clad PCFs are typically used to provide a large mode area for high-power fiber lasers. The primary role of the outer cladding of a double-clad fiber in high-power fiber lasers and amplifiers is to achieve thermal degradation. Because it is a rod-shaped fiber, when targeting a large mode area PCF, one does not need to consider the fiber loss caused by bending, but rather the impact of thermal damage on fiber performance in high-power fiber lasers. With the help of different structures of PCFs, large mode field diameter fibers with single-mode operation can be realized, which is highly essential for developing high-power fiber lasers. PCFs are also highly birefringent. The polarization control device in the laser system can be simplified [85] by setting up a polarization-maintaining unit in the PCFs to ensure a large mode area, single-mode laser output.

According to the inducing factors, the birefringence of optical fibers can be classified into stress-induced birefringence and geometry-induced birefringence. Stress-induced birefringence is due to the different doping in the core and cladding and resultant difference in thermal expansion coefficients of the materials. In the fiber drawing process, different thermal stresses are generated due to different thermal expansion coefficients of the core and cladding, which leads to the anisotropy of the core material and stress birefringence. In contrast to stress-induced birefringence, geometry-induced birefringence (so called intrinsic birefringence) is related to the geometric structure of the core. This noncircular structure of the core may be due to the original structural design of the fiber, or the core may become elliptical from a circle due to the fiber drawing process. In contrast to circularly symmetric fibers, the stresses generated vary largely along the long and short axes due to the asymmetry of the core structure. As a result, the anisotropic glass becomes anisotropic, with a consequent change in the refractive index along the major axes (i.e., n_x_ and n_y_). Optical fibers exhibit intrinsic birefringence as a result of their internal anisotropy, and the higher the core ellipticity is, the higher the value of birefringence.

The main ways to introduce birefringence can be realized through the following ways depicted in Figure 10. First, structural birefringence can be introduced by adjusting the size and shape of the air holes around the core or adjusting the core shape to change the uniform distribution of the refractive index in both directions, i.e., by reducing the symmetry of the PCFs. This method can introduce higher birefringence in PCFs when the initial mode field diameter is small. Ortigosa-Blanch developed the world’s first birefringent PCF [41] at the University of Bath in 2000, for which a high birefringence value of 3.7 × 10^−3^ at 1550 nm was achieved with a mode field diameter of only 3.8 μm. The development of a highly birefringent large mode area silica all-glass fiber (HB-LMA) was reported in 2021 by Warsaw University in Poland. The core of this fiber has a regular nanostructure inside. This microstructure makes the fiber anisotropic. By optimizing the germanium and fluorine doping levels of silica in the core and cladding, a birefringence value of 1.92 × 10^−4^ was obtained in a fiber with a core diameter of 30 µm and an effective mode area of 573 µm^2^, respectively. The same method was used to design a single-mode fiber with a core diameter of 40 µm and a mode field area > 1000 µm^2^ [75]. The structural birefringence value decreases rapidly as the mode field area increases. Therefore, stress birefringence needs to be introduced to further increase the birefringence, which generally induces stress parts (SAPs) with different thermal expansion coefficients from the core inside the fiber.

The largest mode field area of a single-mode polarization-maintaining PCF was reported jointly by the University of Jena, Germany, and Crystal Fiber and A/S, Denmark [62]. The fiber structure is shown in Figure 11a. The fiber has a mode field diameter of 54 µm. Figure 11b shows the near-field intensity distributions of the slow (upstream) and fast axes (downstream) at four different wavelengths. It is suitable for use in high peak power, high-energy fiber lasers, and amplifiers.

For polarization-maintaining PCFs, conventional Panda-type polarization-maintaining fibers can be combined with PCFs. High birefringence is achieved by introducing a stress part (SAP) into the fiber. Multiple combinations of the above methods can also be used to design new structures of highly birefringent PCFs. For example, introducing large air holes around the core of elliptical hole PCFs to increase the birefringence or introducing structures with different thermal expansion coefficients (stress birefringence) while changing the symmetry of the core structure (structural birefringence).

Compared with conventional optical fibers, the excellent optical properties of photonic crystal fibers [27,63,86,87] cause them to have a wide range of promising applications in fiber lasers and amplifiers, fiber sensors, supercontinuous spectrum light sources, and fiber optic communications. Photonic crystal fibers also play a very important role in the laser, communication, and life science fields [88].

Since the first low-loss optical fiber was developed successfully, fiber optic communication has gradually become the main transmission medium of modern communication networks. With the increasing demand for information transmission, the requirements for fiber optic communication systems have evolved toward a very large capacity, ultra-high-speed, ultra-high bandwidth, and ultra-long distance transmission. The capacity of traditional optical fiber communication systems is limited due to the problems of loss, dispersion and nonlinear effects. PCFs are a new type of optical device, and the optical transmission medium brings new vitality to fiber optic communication.

The development of sensing technology of PCFs makes them more sensitive and accurate than traditional optical fibers for sensing and measuring parameters such as the temperature, strain, pressure, refractive index, etc. This can be used to develop various types of sensors for gases, liquids, biology, and pressure, etc. In 2016, N. Ayyanar et al. developed a new hydrostatic pressure sensor using a highly birefringent photonic crystal fiber. A pressure sensitivity of 87.5 pm/MPa was generated through its transmission spectrum [89]. Hoo, Y.L. et al. of the Hong Kong Polytechnic University filled a PBG-PCF with gas and determined the concentration of the gas by monitoring the attenuation of the light passing through the gas. Since the light in the PBG-PCF is transmitted in air, it has higher sensitivity [90]. There are many types of PCF with many unique optical properties, which means that PCF’s sensing elements have many characteristics unmatched by traditional optical fibers, and they are increasingly being used in fiber optic sensing.

Highly nonlinear TIR-PCFs can be used to generate supercontinuous spectra. By varying the PCF structure, flexible and adjustable dispersion characteristics, cutoff-free single-mode characteristics, and high nonlinear coefficients [91] can be obtained, resulting in wide and flat SC. Ebnali-Heidari et al. achieved low-loss, ultra-flat dispersion by filling the PCF with an optical fluid to control the dispersion. The superflat supercontinuum spectrum of 640~1180 nm was found to be achieved by simulations with femtosecond pulses centered at 1250~1625 nm [92]. The dispersion of ordinary passive fibers and rare earth doped fibers cannot be flexibly adjusted as required. Additionally, it is not easy to produce a wavelength component shorter than the central wavelength of the pump light.

The first PCF laser was developed at the University of Bath, UK, in 2000. Using a titanium gem laser (970 nm) pumping a Yb^3+^-doped photonic crystal fiber which was 81 mm long, a laser output of 1040 nm was obtained [93]. In the research and application of high-power fiber lasers, PCFs have gradually developed into one of the research hot spots in the field of fiber lasers due to their many excellent characteristics. PCFs can be used in lasers not only as a gain medium directly, but also as a dispersion compensation and transmission of optical energy. By optimizing the structure of a PCF, the output power of the fiber laser can be increased, the beam quality of the fiber output can be optimized, and the system of the fiber laser can be simplified.

## 5. Conclusions

The concept, classification, and light-guiding mechanism of PCFs are first introduced in this review, followed by the theoretical analysis method and properties of PCFs, and a brief review of the development of PCFs with increasing mode area and tailored polarization-maintaining properties with a global scope. On this basis, this review highlights several of the main technical paths and design methods to achieve a large-mode field and polarization-maintaining property. With the rapidly growing demand for high-speed, high-capacity information transmission and high-power fiberoptics in industry, the research on large-mode area PCFs is in high demand and becoming more extensive, with which has come the emergence of new LMA-PCFs with different microstructures. The peculiar optical properties of PCFs make it possible to overcome many difficulties faced by traditional fiber lasers. PCFs are widely used in the fields of high-power fiber lasers and amplifiers, high-speed information transmission, high-power energy transmission, high-sensitivity sensors, supercontinuum light sources, deflectors, fiber grating and dense WDM systems.

However, there are still some problems to overcome: (1) For the improved refractive index guided PCFs, the fiber core replacing one or more layers of air hole ring can significantly increase the mode field area, but the single-mode cutoff wavelength is affected by large fluctuations. There is no unified theoretical analysis to study the effect of PCFs as the core diameter increases and the different normalization parameters of the fiber change on the single-mode property. (2) The current theoretical simulations have been used on many new structures of LMA-PCFs, but most of the designed structures are too complex and specialist in nature, as they have very small refractive index variation differences. In the real fiber design process, it is difficult to realize the precise structural control of PCFs and so it is hard to successfully design PCFs with a complex structure but very uniform refractive index distribution. (3) The current PCF products that achieve mass production are generally too expensive, and they are mostly used only in the laboratory and cannot be more universally applied in industrial fields. (4) To strengthen PCF theoretical simulation analysis and optimize the fabrication process, studies need to integrate the fundamental research with PCFs’ potential applications in a better way. It is believed that as the research progresses, the gap between theoretical simulation and fiber fabrication will be gradually narrowed, and the LMA-PCFs will have more broad application potential. 

## Figures and Tables

**Figure 1 materials-15-01558-f001:**
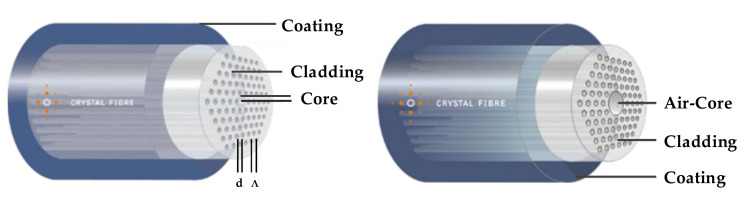
Microstructural arrangement of solid-core PCFs and hollow-core PCFs [7].

**Figure 2 materials-15-01558-f002:**
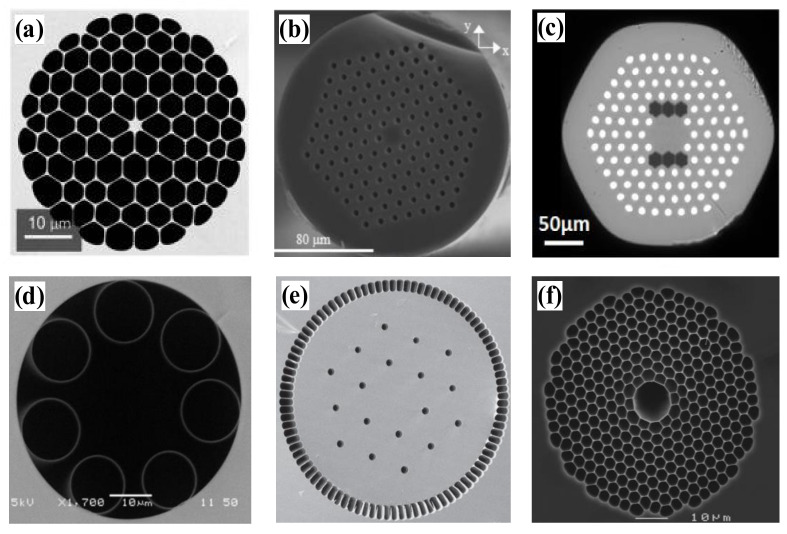
Microstructural arrangement of PCFs. (**a**) Total internal reflection (TIR) [11]; (**b**) Total internal reflection (TIR) [12]; (**c**) Polarization-Maintaining Photonic Crystal Fiber [13]; (**d**) Hollow-core anti-resonant (HC-AR) PCFs [14]; (**e**) Solid-core large-pitch photonic crystal fiber [15]; (**f**) Hollow-core photonic bandgap fibers (HC-PBGFs) [16].

**Figure 3 materials-15-01558-f003:**
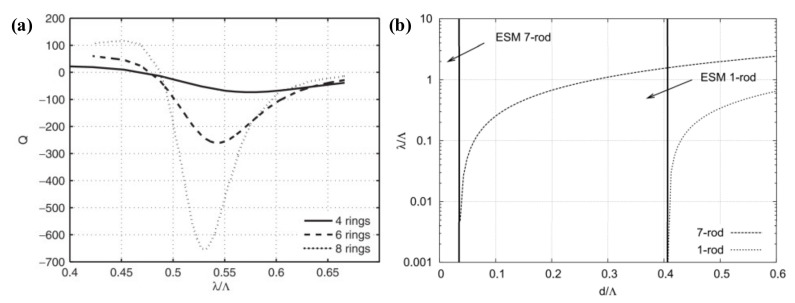
(**a**) Variation in the Q value with normalized wavelength λ*/Λ as the number of air hole rings varies [36]; (**b**) cutoff wavelength of fiber cores replacing 1 and 7 air hole PCFs [6].

**Figure 4 materials-15-01558-f004:**
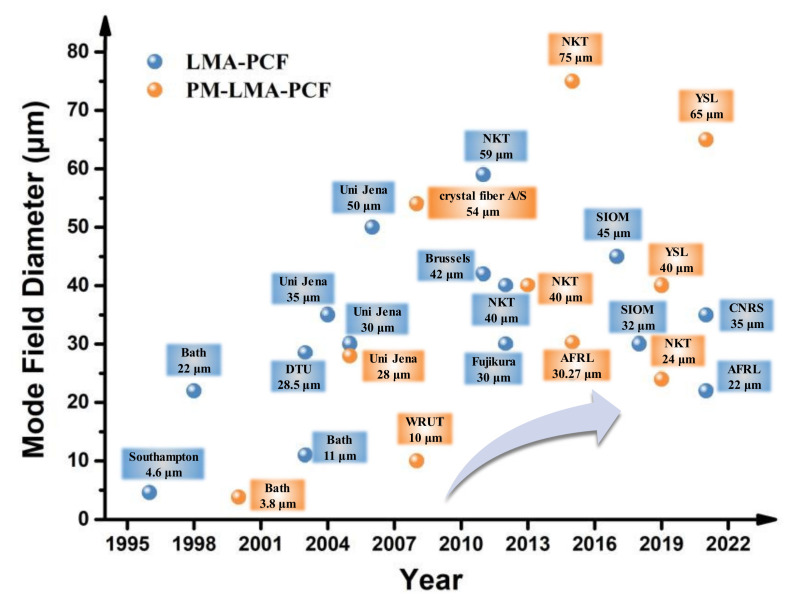
The landmark research advances in single-mode (SM) LMA-PCFs and polarization-maintaining (PM) LMA-PCFs from 1996 to 2021.

**Figure 5 materials-15-01558-f005:**
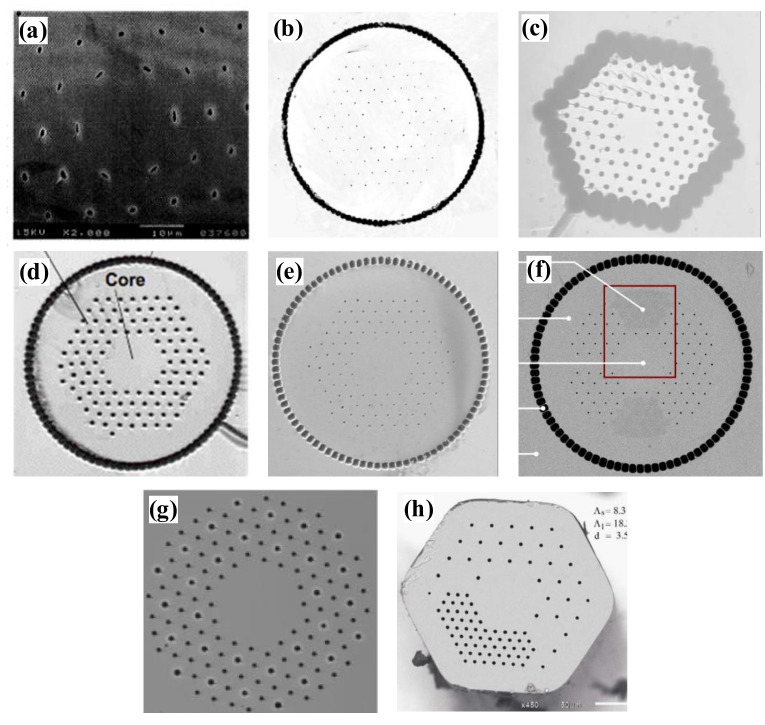
Schematic structure of LMA-PCFs (**a**) 22.5 μm core diameter developed at Bath University [51]; (**b**) 35 µm mode field diameter rod PCF designed at Jena University [55]; (**c**) Yb-doped PCF with a mode field diameter of 30 μm developed at Jena University [54]; (**d**) 100 μm Yb-doped rod PCF developed at the AFRL [67]; (**e**) 50 μm mode field diameter developed at Jena University [61]; (**f**) 54 μm mode field diameter reported by Crystal Fiber A/S [62]; (**g**) 59 μm mode field diameter PCF developed by NKT [64]; (**h**) 42 μm mode field diameter reported by the University of Brussels [59].

**Figure 6 materials-15-01558-f006:**
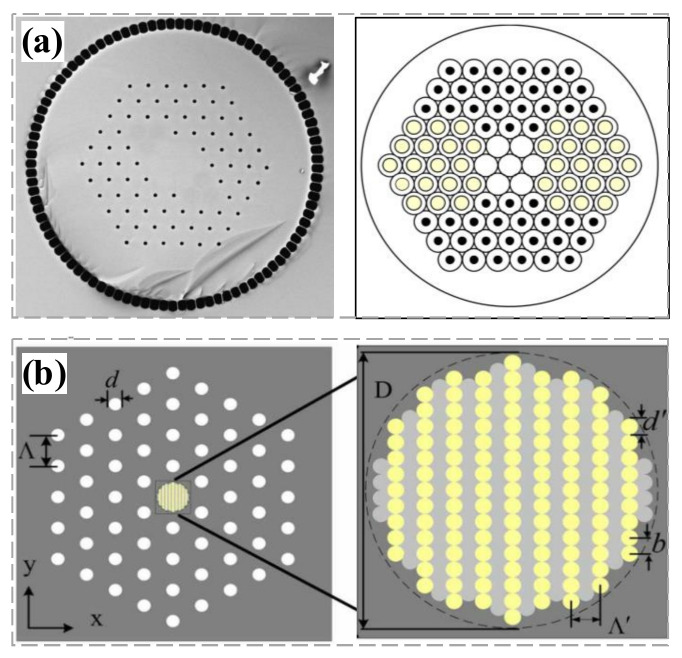
Fiber cross-sections and design structure. (**a**) A single-polarization single transverse mode large mode area PCF developed at the University of Jena, Germany, 2005 [52]; (**b**) A new highly birefringent large-mode area PCF proposed by Ming-Yang Chen at Jiangsu University, 2007 [68].

**Figure 7 materials-15-01558-f007:**
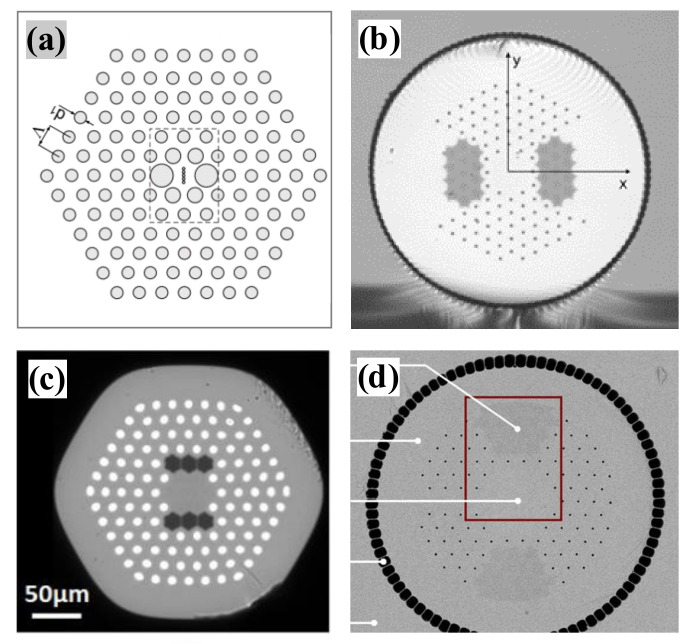
(**a**) (2008) Wroclaw University of Technology introduced small holes in the central region of the core to induce birefringence; (**b**) (2013) NKT fabricated fiber named DC-200/40-PZ-Yb [58]; (**c**) (2015) Clemson University developed single-polarized, single-mode operating Yb-polarized all-solid-state photonic bandgap fiber [11]; (**d**) (2008) 54 μm mode field diameter reported by Crystal Fiber A/S [62].

**Figure 8 materials-15-01558-f008:**
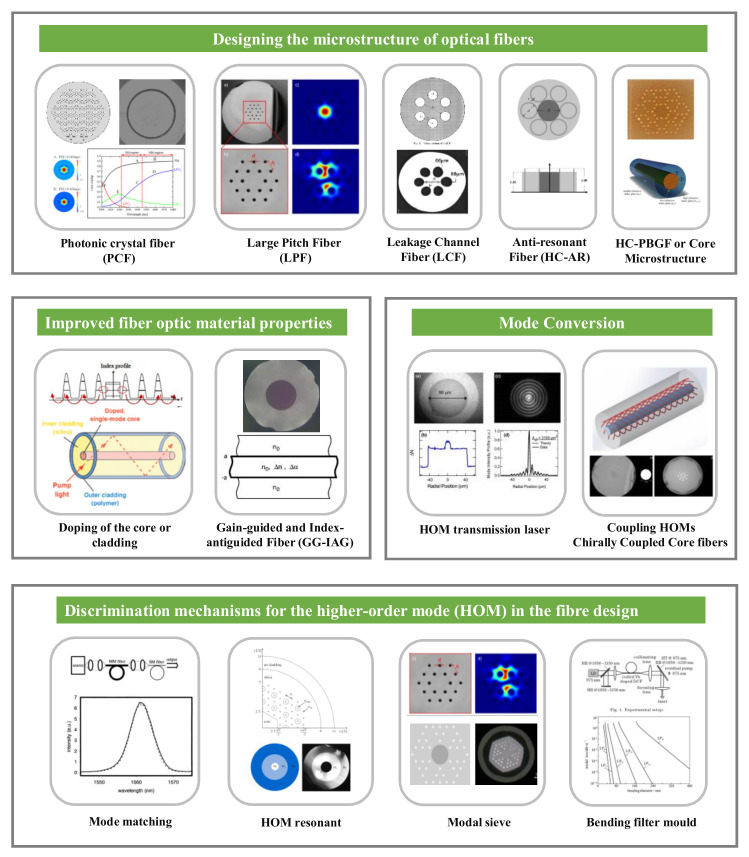
Technical routes [11,39,58,64,71,72,73,74,75,76,77,78,79,80,81] to achieve single-mode large field area for PCFs.

**Figure 9 materials-15-01558-f009:**
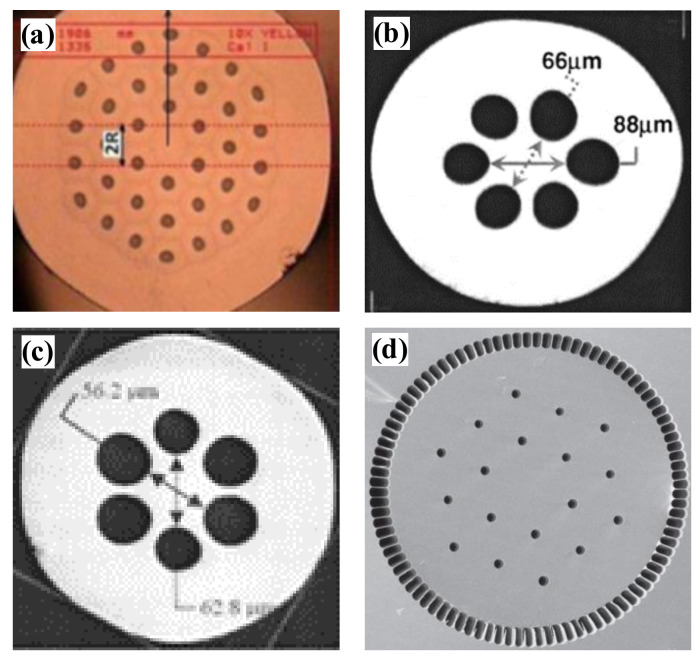
(**a**) All-solid large mode area (LMA) fiber [51] reported by IMRA, 2009. (**b**) LCF for passive leaky channel fiber designed by IMRA, 2007 [73]; (**c**) LCF for Yb-doped leaky channel fiber designed by IMRA, 2007 [73]; (**d**) large-pitch photonic crystal fiber [12].

**Figure 10 materials-15-01558-f010:**
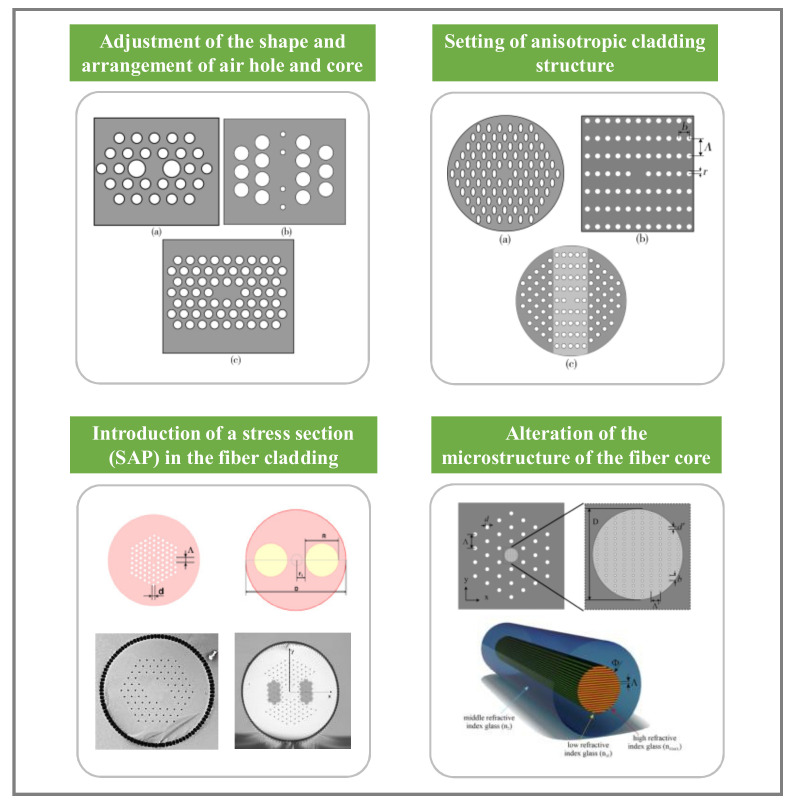
Technical route to achieve LMA-PCFs with high birefringence value [52,58,62,75].

**Figure 11 materials-15-01558-f011:**
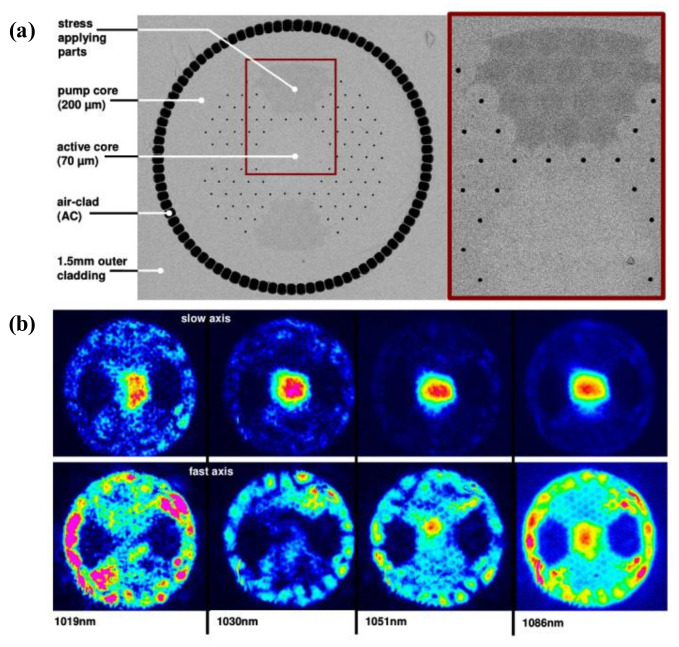
(**a**) Schematic diagram of the fiber structure (**b**) near-field intensity distribution of the slow (upstream) and fast axes at four different wavelengths [62].

**Table 1 materials-15-01558-t001:** Specific parameters of the single-mode LMA-PCFs and PM-LMA-PCFs compared in Figure 4.

Year	Fiber Structure Parameters Core Diameter (D_core_) Cladding Diameter (D_clad_) Air Hole Diameter (d_air_) Pitch (Λ)	Fiber Structure Schematic	Mold Field Diameter and PM Performance	Countries and Institutions	Refs.
2000	D_clad_ = 63 µm, Λ = 1.96 µm the small holes, d_1_ = 0.40 µm the large holes, d_2_ = 1.16 µm	/	PM-LMA-PCF MFD = 3.8 μm B = 3.7 × 10^−^^3^	University of Bath, UK	[41]
1996	D_core_ = 4.6 µm, Λ = 2.3 µm, D = 0.2–1.2 µm (8 air hole rings).	/	MFD = 4.6 μm	University of Southampton, UK	[48]
2003	D_core_ = 7.5 µm (Yb^3+^-doped) Structure 1: d/Λ = 0.30, Structure 2: d/Λ = 0.55	/	MFD = 11 μm	University of Bath, UK	[50]
1998	D_core_ = 22.5 µm, Λ = 9.7 µm, d = 1.2 µm	Figure 5a	MFD = 22 μm	University of Bath, UK	[51]
2005	Λ = 12.3 µm, d_air_/Λ = 0.09, d_BS_/Λ = 0.25, α_BS_ = 5·10^−^^7^/K, α_FS_ = 10·10^−^^7^/K	Figure 6a	PM-LMA-PCF MFD = 28 μm PER = 15.5 dB	University of Jena, Germany	[52]
2012	D_core_ = 48 μm, (7-cell defect core). Λ = 12 µm, d/Λ = 0.17.	/	MFD = 30 μm	Optics and Electronics Laboratory, Fujikura. Japan.	[53]
2005	D_core_ = 35 μm, d/Λ = 0.33, D_clad_ = 117–141 μm,	Figure 5c	MFD = 30 μm	University of Jena, Germany	[54]
2004	D_core_ = 40 µm, D_clad_ = 170 µm, NA = 0.03 d = 1.1µm, Λ = 12.3 µm (4 air hole rings)	Figure 5b	MFD = 35 μm	University of Jena, Germany	[55]
2021	D_core_ = 40 μm, (19-cell defect core) Λ = 9.4 µm, d/Λ = 0.8.	/	PM-LMA-PCF MFD = 35 μm PER = 17 dB	French National Center for Scientific Research (CNRS)	[56]
2015	Three low index B-doped silica stress rods on both sides of the fiber core	Figure 7c	MFD = 38 μm PER = 21 dB	Clemson University, Air Force Research Laboratory, US	[11]
2012	DC-200/40-PZ-Yb	Figure 7b	PM-LMA-PCF MFD = 40 μm PER = 25~30 dB	NKT Photonics, Denmark	[57] [58]
2011	Λ_S_ = 8.5 µm, Λ_1_ = 17 µm, d = 3.94 µm	Figure 5h	MFD = 42 μm	Vrije Universiteit Brussel, Belgium	[59]
2017	D_core_ = 50 μm, D_clad_ = 260 μm d = 2.5 µm, Λ = 20 µm	/	MFD = 50 μm	Shanghai Institute of Optics and Fine Mechanics (SIOM)	[60]
2006	D_core_ = 60 μm, (19-cell defect core). d/Λ = 0.19 (4 air hole rings).	Figure 5e	MFD = 50 μm	University of Jena, Germany	[61]
2008	D_clad_ = 200 μm, D_core_ = 70 μm, NA = 0.6, (19-cell defect core).	Figure 5f	PM-LMA-PCF MFD = 54 μm polarization better than 85%	Crystal Fibre A/S, Denmark	[62]
2018	D_core_ = 58 μm, D_core_ = 79 μm	/	PM-LMA-PCF MFD = 45/58 μm B = 3.54 × 10^−^^5^	University of Limoges, CNRS, France	[27] [63]
2011	D_core_ = 85 µm (Yb^3+^-doped), Λ = 14.5 µm, 0.1 < d/Λ < 0.3	Figure 5g	MFD = 59 μm	NKT Photonics, Denmark	[64]
2022	/	/	PM-LMA-PCF MFD = 65 μm PER = 18 dB	Yangtze Soton Laser CO. (OYSL), China	[65]
2015	DC-285/100-PM-Yb	/	PM-LMA-PCF MFD = 75 μm	NKT Photonics, Denmark	[66]

## Data Availability

The data presented in this study are available upon request from the corresponding author.

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
