# Peer review of "Advances in Silica-Based Large Mode Area and Polarization-Maintaining Photonic Crystal Fiber Research"

_materials, 2022, doi:10.3390/ma15041558_

Round 1
Reviewer 1 Report
Since this is review paper, more discussions needed in the concept of PCF. Please find my detailed comments.
Comments on materials-1559564 “Advances in silica-based large mode area and polarization-maintaining photonic crystal fiber research”
In this work, photonic crystal fiber (PCF) and its large mode area and polarization-maintaining properties are reviewed and presented. Before considering for publication, few concerns are strongly suggested to carry out as given below.
- Many features of PCFs are investigated by numerous researchers. Then why authors mainly focused only on PCF with LMA and Polarization maintaining PCF. Justify.
- Give units for equations where it is applicable.
- Authors must include some applications of PCF like Terahertz, sensing and supercontinuum generation.
- References are inadequate in the manuscript.
- Abstract must be revised according to novelty of paper.
- For the applications, authors may provide comparison table.
- Authors have not focused on applications point of view which is essential for review paper.
- Could you explain in detail about the fabrication process of PCF? And steps involved in designing?
- Authors present various structures of PCF without its significance. Kindly include their features.
- Experimental feasibility of PCF structures must be discussed.
This manuscript must be improved in all aspects before the consideration for publication.
The following papers must be cited when authors investigate about PCF characteristics
- https://ieeexplore.ieee.org/document/7782834
- https://ieeexplore.ieee.org/document/8758322
Reviewer 2 Report
The authors are giving a thorough review on silica-based photonic crystal fibers and related design and manufacturing techniques. As a perspective paper, the given summary would be useful for the scholars and the industrial community.
The work is well organized and described. The introduction provides sufficient background and the methods adequately described and well supported with relevant references. The general concept is clearly presented and well explained with the supplementary materials. Regarding the scientific soundness and novelty are not applicable in this type of a review paper since it is a summary of already well-studied and previously published works.
The paper is already complete, yet some minor points need to be adjusted. The English language and style are understandable, yet an overall spell check is required. All figures needs to be re-arranged in terms of the font and the format to keep the unity throughout the text. Moreover figure quality must be improved.
I would like to thank all the Authors for their efforts and I kindly ask them to address my comments and suggestions. I would recommend publication, once the minor issues are adjusted.
Author Response
We would like to thank you very much for your valuable comments and good suggestions that greatly helped to improve our manuscript. Thank you very much for your time and efforts. We are sorry for the trouble caused by some spelling and grammar mistakes in this manuscript. We have asked for MDPI Language Service to recheck and edit this article and corrected these mistakes.
Thank you again for your careful review and pertinent comments on our manuscript. We marked all the changes using the review mode in the revised manuscript.
Reviewer 3 Report
- The title dictates review article but I am not satisfied with past studies reported in this paper, I suggest authors to add more references to advocate your findings. I am copying here one useful link: https://doi.org/10.1039/D1RA07021E, etc etc..
- Language editing is required.
- In ‘Abstract’ section, no need to add these sentences (line 10-14) instead should covers in one sentence
- Section 2 (all sub-headings) needs to be concise and should includes concern literature
- Figure 3 is not visible, need higher resolution
- d/Λ stands for what?
- In fig.3, authors explained Q-parameters with regards to ref.[30] but in figure.3 they used ref[6]?why
- Put eq.5 in 4 and then explain Core overlap factor
- Fig 4 in tabular or histogram should be more appealing
- In section.4 there needs to put ref for technical aspects of optical fiber application such as: https://doi.org/10.1364/OPEX.12.000956, https://doi.org/10.1155/2013/148903 , etc etc..
- Conclusion section needs to be concise.
Round 2
Reviewer 1 Report
Dear Authors,
Revised manuscript is suitable for publication
Reviewer 3 Report
All my concerns have been addressed and i would like to recommend the article in this form.
Just a quick note for authors:(for future)
Whenever you add ref's(as asked or else) it is required to highlight them too in the revised version so as to see exactly.